# A Sequential Path Model Testing: Emotional Intelligence, Resilient Coping and Self-Esteem as Predictors of Depressive Symptoms during Unemployment

**DOI:** 10.3390/ijerph18020697

**Published:** 2021-01-15

**Authors:** María Angeles Peláez-Fernández, Lourdes Rey, Natalio Extremera

**Affiliations:** 1Department of Social Psychology, Social Work, Social Anthropology and East Asian Studies, University of Málaga, 29071 Málaga, Spain; nextremera@uma.es; 2Department of Personality, Assessment and Psychological Treatment, University of Málaga, 29071 Málaga, Spain; lrey@uma.es

**Keywords:** unemployment, emotional intelligence, resilience, self-esteem, depression

## Abstract

The unemployment rate has dramatically increased in southern Europe in the last decade. Although it is well-known that unemployment impairs mental health, the specific roles of personal resources like emotional intelligence (EI) and potential underlying mechanisms remain unclear. Prior studies have shown that resilience and self-esteem are mediators in the link between EI and mental health. The present study aims to bridge these gaps by testing a sequential path model. Specifically, we propose that EI is associated with lower depressive symptoms, which is explained by higher resilient coping strategies and a resulting increased self-esteem among unemployed individuals. A sample of Spanish unemployed persons completed measures of EI, resilience, self-esteem and depression. The results showed that higher levels of EI were positively associated with resilience and self-esteem and negatively related to depressive symptoms. Path analyses showed that resilience and self-esteem mediated the relation between EI and depression in sequence. These findings suggest that EI plays a key role in promoting mental health and provide preliminary evidence regarding potential mechanisms through which EI contributes to mental health during unemployment. Implications for assessing the absence of these positive resources in developing effective job search programs geared toward promoting mental health and re-employment are discussed.

## 1. Introduction

The unemployment rate has dramatically increased in southern European countries in the last two decades, especially after the global economic crisis, particularly affecting Greece (16.8%), Spain (16.2%) and Italy (9.8%), which manifestly exceed the EU average rate (8.4%) [Eurostat, October 2020] [1]. Unemployment is a psychosocial stressor with deleterious and long-term effects on physical and mental health, including lower perceived well-being and quality of life [2], reduced life satisfaction [3] and higher rates of suicide, anxiety and depression [4]. Specifically, there is empirical evidence that mental health problems such as depressive symptoms are more prevalent for the unemployed; also, the highest odds ratios for depressive symptoms are reported in those who moved from permanent employment to unemployment [5]. The increase in depression among unemployed people might be associated with elevated psychological distress, financial insecurity and the loss of a time structure, employment benefits and social status [6,7]. Unemployment-related mental health problems and chronic diseases impair unemployed individuals’ ability to work and increase the likelihood of drifting into long-term unemployment as a result of being laid off or of ineffective attempts at re-employment [8,9]. Therefore, career advisors and social researchers need a broader understanding of the underlying mechanisms involved in the process of coping with job loss and need to identify protective factors that help unemployed people to manage negative consequences of unemployment, such as increased distress and reduced psychological well-being [7].

Among these potential protective factors, emotional intelligence (EI) is remarkable. From an ability perspective, EI is a psychological construct comprised of some basic emotional skills: emotion perception, assimilation and understanding and emotional regulation [10]. Numerous studies have provided evidence to suggest that these emotional abilities are key factors of physical and mental health correlates [11,12]. In short, there is a great deal of empirical and meta-analytic research supporting a significant and positive association between EI and improved mental health, interpersonal functioning and subjective well-being [13,14,15], lower emotional distress [16,17], higher life satisfaction [18,19,20,21], greater self-esteem [20], higher social support [19,22], greater positive affect and lower negative affect [16,18,19], better subjective well-being and fewer psychological symptoms [13]. Specifically, there is general consensus, across different clinical and nonclinical samples and using different methodological approaches to EI measurement, that EI is negatively related to depression, suggesting that EI might be a protective factor against depression [13,23]. Specifically, recent research on unemployment and health has shown that EI plays a key role in reducing negative psychological symptoms associated with unemployment, including depression, and in boosting well-being [24,25,26,27,28,29].

Although prior research literature is in agreement that emotional skills are associated with lower depressive symptoms, the potential mechanisms involved in this relationship are less known. Research on EI suggests several mechanisms that might explain this link [15]. One of the general mechanisms that may relate EI to lower depressive symptoms is resilient coping. Resilient coping has been conceptualized as a dynamic process that includes a positive adaptation and a tendency to cope in a highly adaptive manner within the context of significant stress or adversity [30]. There is theoretical and empirical support showing that individuals with high levels of EI are more likely to implement resilient coping strategies to manage negative emotions, as well as to use them more effectively in dealing with social stressors [31,32,33,34,35,36], which in turn reduces negative affects related with mental health disorders such as major depression [37]. In addition, some studies have shown that resilient coping also shows a mediating role between emotional intelligence and some psychological symptoms or well-being outcomes in both youth and adult samples. For example, adaptive coping was found to mediate the effects of EI on mental health in gifted students [38]; resilient coping partially mediated the link trait EI and aggressive and rule-breaking behaviors among adolescents [39]; resilience mediated the effects of EI on exam anxiety and academic stress among university students [40]; and coping strategies mediated the relationship between trait EI and occupational stress among health care professionals [41].

Another potential candidate that may mediate the association between EI and depression is self-esteem. Self-esteem refers to a global feeling of self-worth or adequacy as a person or to generalized feelings of self-acceptance, goodness and self-respect [42]. Empirical research shows that EI significantly predicts higher levels of self-esteem [20,43]. Also, negative feelings of self-worth predict higher levels of depression [44]. Additionally, there is a substantial body of research demonstrating that EI is associated with higher well-being and life satisfaction both directly and indirectly through self-esteem [20,45]. Finally, a recent study has provided some empirical evidences for the mediating role of self-esteem in the EI–depression link among university students [46].

Therefore, there is empirical support for the association between EI with resilient coping and self-esteem and between resilient coping and self-esteem with lower levels of depression; hence, the assessment of resilient coping and self-esteem as potential mediators of the association between EI and depression seems to be warranted. In fact, resilient coping and self-esteem have been consistently found to play a relevant role between EI and depressive symptoms. However, most of these studies only examined the role of a single mediator in the EI-depression linkage and have rarely evaluated effects of the mediators concurrently. Thus, to the best of our knowledge, no prior research has examined this cumulative effect and compared it with the independent mediating effects of resilient coping and self-esteem among the unemployed population. Exploring these serial mediating effects among the unemployed would elucidate key factors and mechanisms of mediation related to the unemployed population and would help career counselors to consider how to alleviate mental health problems more efficiently and to specifically target the needs of unemployed population. Therefore, the purpose of the present study was twofold: First, we sought to examine the relations between EI, resilient coping, self-esteem and depressive symptoms among Spanish unemployed individuals. Second, we sought to determine whether resilience and self-esteem mediated the relation between EI and depression in sequence. Since prior studies have shown that resilient individuals are more likely to have a positive sense of themselves and to regard themselves highly [47,48], we expected that both mediators might act in this sequence; that is, individuals using resilient coping would enhance the development of positive cognitions about themselves which, in turn, would lead them to an amelioration of depression. Overall, considering prior research on the significant associations between EI, resilience, self-esteem and depressive symptoms along with the critical effects of resilient coping and self-esteem on mental health, we stated the following research hypotheses:
**Hypothesis** **1.***EI is positively associated with higher self-esteem and resilient coping and negatively linked to depressive symptoms*.
**Hypothesis** **2.**(Single mediation). *EI predicts higher levels of resilient coping and self-esteem. These variables, in turn, independently predict lower levels of depression*.
**Hypothesis** **3.**(Sequential mediation). Resilient coping and self-esteem might serve as mediators in a sequential mediation model between EI and depression; that is, *EI positively predicts resilient coping, leading to stronger self-esteem, further decreasing depression*.

## 2. Materials and Methods 

### 2.1. Participants and Procedure

The sample comprised 530 Spanish unemployed persons (61.1% women; 38.9% men) who volunteered in a research on “unemployment and well-being” in diverse National Employment agencies in the south of Spain. The individuals visiting the centers were contacted by a qualified career guidance professional working in an employment agency and were invited to respond to a survey anonymously.

Altogether, the mean age of the sample was 34.60 years (CI 95%: 33.64–35.57; range 16 to 64 years). The level of education was: 9.2% without studies; 40.6% primary studies; 18.2% uncompleted secondary studies; 15.9% completed secondary studies; 13.8% university studies; 1.9% postgraduate studies. The average length of unemployment was 22.80 months (CI 95%: 20, 30–25, 31; SD = 26.92 months). Inclusion criteria were being unemployed and in an active job search. About 70% of those invited were disposed to participate. The study protocol was conducted in compliance with the Declaration of Helsinki and endorsed by the Research Ethics Committee of the University of Málaga (66-2018-H).

### 2.2. Measures

#### 2.2.1. Emotional Intelligence

We used the Spanish version of the Wong and Law Emotional Intelligence Scale (WLEIS) [49] to measure self-reported EI. The WLEIS includes 16 items (e.g., “*I am sensitive to the feelings and emotions of others*”; “*I can always calm down quickly when I am very angry*”) to be responded to on a seven-point scale ranging from 1 (strongly disagree) to 7 (strongly agree). We then calculated a global EI score where higher scores indicated a greater EI. The WLEIS has shown high levels of reliability and validity in Spanish samples [50]. In this study, the reliability coefficient was 0.92.

#### 2.2.2. Resilient Coping

Resilient coping was measured by the Brief Resilient Coping Scale (BRCS) [30], which assesses adaptability to stressful circumstances. The BRCS includes four items that reflect cognitive and behavioral coping strategies: “*I look for creative ways to alter difficult situations*”; “*Regardless of what happens to me, I believe I can control my reaction to it*”; “*I believe I can grow in positive ways by dealing with difficult situations*”; and “*I actively look for ways to replace the losses I encounter in life*”. Participants were asked to rate the items in response to the following statement: “Consider how well the following statements describe your behavior and actions on a scale from 1 to 5, where 1 means the statement does not describe you at all and 5 means it describes you very well”. Item ratings were summed to yield a total score ranging from 5 to 20. In the current study, Cronbach’s alpha was 0.75 for this four-item scale. High scores on the BRCS denote a flexible, optimistic, creative, determined coping style that includes a tendency to reframe losses into challenges that can be overcome. Both English and Spanish versions have shown evidence of discriminant validity and appropriate internal consistency [30,51]. Cronbach’s alpha in this study was 0.81.

#### 2.2.3. Self-Esteem 

Self-esteem was measured using the Single-Item Self-Esteem (SISE; [52]). Participants rated the item (“*I see myself as someone who has high self-esteem*”) on a five-point Likert scale ranging from 1 (disagree strongly) to 5 (agree strongly). The SISE has been found to have high test–retest reliability, criterion validity coefficients above 0.80 on the Rosenberg Self-Esteem Scale (RSE) and a similar pattern of construct validity coefficients as the RSE with 37 different constructs [52]. Using longitudinal data, the reliability of the SISE was 0.75 [52]. This single item was translated from English into Spanish using the method of back-translation.

#### 2.2.4. Depressive Symptoms

Depression was measured by the depression subscale from the 21-item Depression, Anxiety and Stress Scale (DASS-21; [53]), which consists of seven items on a Likert-type scale, designed to measure the negative emotional states of depression in the past week, where “0 = did not apply to me at all” and “3 = applied to me very much, or most of the time” (e.g., “*I couldn’t seem to experience any positive feelings at all*”). The Spanish version showed satisfactory internal consistency and adequate divergent and convergent validity [54]. Cronbach’s alpha in this study was 0.91. 

## 3. Data Analysis

The statistical program SPSS version 25 (IBM, Armonk, NY, USA) was used to perform the descriptive statistical analyses, reliability analyses and bivariate correlations. To assess the possible bias arising from common method variance, this study followed Harman’s single-factor technique as suggested by Podsakoff et al. [55]. Thus, in order to test for sequential mediation effects, Hayes’s SPSS macro PROCESS (Model 6) with a 95% bias-corrected confidence interval (CI) based on 5000 bootstrap samples was used [56]. Bootstrapping is an approach that resamples the original sample size from the data multiple times and does not rely on the assumption that data are normally distributed. Bootstrapping analysis provides the most powerful and reasonable method of obtaining confidence limits for specific indirect effects as it is free from assumptions regarding the shape of the sampling distribution of the indirect effect and also has better control for type I errors. This procedure allows for the examination and statistical testing of each of the estimated indirect (e.g., mediated) effects in a model and the direct effect of the independent variable on the criterion variable, while controlling for effects of potential serial mediators. Then, using the estimates on the basis of these 5000 bootstrap samples, the mean direct and indirect effects and their confidence intervals (CIs) were calculated. These CIs were used to determine whether each effect (e.g., direct, indirect) was statistically significant. For each effect, we examined the 95% CI and if the value of 0 did not fall within the range of the CI for that effect, then the finding was statistically significant. The serial mediation process estimated three specific indirect effects of emotional intelligence on depressive symptoms: (a) the indirect path via resilient coping; (b) the indirect path via self-esteem; and (c) the indirect path via resilient coping to self-esteem. 

### 3.1. Descriptive Analyses

Means, standard deviations, reliabilities and Pearson’s correlations among the evaluated variables are presented in Table 1. As can be seen, EI was negatively and significantly associated with depressive symptoms and positively and significantly related to resilient coping and self-esteem. Thus, resilient coping and self-esteem were significantly and positively related to each other. Finally, resilient coping and self-esteem were negatively and significantly associated with depressive symptoms. Harman’s Single Factor analysis identified 23 factors with eigenvalues > 1.00 and the variance explained by a single factor was found to be 36.18%, which is less than the 50% recommended cut-off criteria [55]. Consequently, these findings suggested that common method variance does not significantly inflate correlations and may not be of great concern and is likely to have minimal impact on the interpretation of the results.

### 3.2. Serial Mediational Analysis

We examined whether the relationship between EI and depression was sequentially mediated by resilient coping and self-esteem. Both gender and age were added as control variables. To further clarify the direction of the indirect effect, we also tested an alternate model in which self-esteem preceded resilient coping. Results of the mediation analyses are presented in Table 2.

As shown in Figure 1, two of the three hypothetical mediating effects were supported. First, the specific indirect effects of EI on depression through resilient coping (independent of self-esteem) were not supported (B = −0.01, SE = 0.03; 95% CI = −0.08, 0.04). Second, self-esteem was found to mediate the association between EI and depressive symptoms (B = −0.05, SE = 0.01; 95% CI = −0.089, −0.029). Third, the sequential pathway of EI→resilient coping→self-esteem→depression was significant (B = −0.03, SE = 0.01; 95% CI = −0.068, −0.015). Accordingly, higher levels of EI were serially associated with higher resilient coping, higher self-esteem and finally lower depressive symptoms. Thus, the residual direct pathway between EI and depression was also significant (b = −0.18, *p* < 0.001). Therefore, resilient coping and self-esteem only partially mediated the link between EI and depression among the unemployed. This final serial mediation model was significant, accounting for 17% of the variance in depressive symptoms (R^2^ adj = 0.17; F (5, 500) = 20.50; *p* < 0.01). Pairwise comparisons between the two significant indirect effects on the EI-depression linkage were conducted to compare the strengths of these associations. No significant difference was observed between the serial mediating effect and the indirect effect through self-esteem.

Interestingly, when substituting the order of the mediators so that self-esteem preceded resilient coping in the model, the indirect serial path was no longer significant (B = −0.01, SE = 0.02; 95% CI = −0.055, 0.032).

## 4. Discussion

The present study investigated whether resilient coping and self-esteem, which prior literature confirmed as relevant variables related to psychological well-being, mediated the relationship between EI and depression in a sample of Spanish unemployed individuals.

The results of path analyses revealed that resilient coping and self-esteem played a sequential mediating role between EI and depression, suggesting that EI was positively linked to higher resilient coping and self-esteem, which in turn predicted lower levels of depressive symptoms. The findings lend support to the notion that EI decreases depressive symptoms indirectly, suggesting that greater resilience and self-esteem may be potential underlying sequential mechanisms through which emotional abilities contribute to reduced depressive symptoms during unemployment. Our identification of two mediators in sequence supports the notion that lacking ability to cope in a highly adaptive manner with the demands of stressful experiences during unemployment may represent a risk factor that leads to reduced positive feelings of self-worth among unemployed individuals, which might, in turn, contribute to depressive symptoms among the unemployed. The mediating effect of resilient coping is consistent with past research reporting the idea that individuals with high EI are more likely to find creative ways to deal with difficult situations [33]. This resilient coping style might increase the individual’s general sense of self-worth [47,48], and, in turn, decrease depressive symptoms [44]. With respect to individual mediation analysis, the specific indirect effects of EI on depression through resilient coping (independent of self-esteem) was not supported, which means that resilient coping did not independently mediate the relationship between EI and depression; while EI had a significant direct effect on resilient coping, resilient coping did not show a significant direct effect on depressive symptoms. This may be due to different methodological and theoretical reasons. Firstly, the lack of a significant effect found in our study is a tentative result due to the limited sample size and further studies should be conducted with larger and heterogenous samples. Also, past studies have shown that resilient coping is associated with better mental health outcomes [34]; however, some researchers have found a specific and unique role for resilience when comparing different internalizing symptoms, with resilience being more closely associated with anxiety than depression symptoms [57]. Each construct includes a unique component, with heightened autonomic arousal being specific to anxiety and low positive affectivity being specific to depression. Accordingly, it is plausible that resilient coping might not play a major role in reducing the core component in depression, being more critical for reducing the central core in anxiety (low autonomic arousal). Finally, our findings also suggest that some self-system processes might mediate the effect of resilient coping on depressive symptoms such as self-esteem. Pending replication, future researchers are advised to examine this issue in depth.

Regarding self-esteem, the specific indirect effects of EI on depression through self-esteem (independent of resilience) were supported, thus supporting the notion that self-esteem independently mediates the relationship between EI and depression. Altogether, self-esteem (rather than resilient coping) had the most significant impact on this association. The results contribute to the current literature on the relationship between EI and depression by providing preliminary evidence that self-esteem plays a more important individual role than resilient coping in the linkage. Moreover, the serial mediation was also significant, suggesting that EI is associated with greater resilient coping and that resilient coping subsequently favors self-esteem, thus predicting lower depressive symptoms. These findings are in accord with prior meta-analytic research corroborating the robust effect of negative feelings of self-worth on depression [44]. Our results are also consistent with past studies on the indirect role that EI plays in mental health and well-being outcomes by means of different individual and social factors [16,34,35,36,38], and contributes to the current literature by extending our understanding of the mechanism that underlies the linkage of EI and depression among the unemployed population.

In addition, these findings provide novel evidence of promising mediating variables that might be employed as tools for counseling and therapeutic interventions. Indeed, some practical implications are derived from this work, especially relevant in the area of designing prevention programs aimed at identifying and eliminating factors that might lead to the development of mental health problems among the unemployed. Although there are many preventive job search programs to promote re-employment and prevent the adverse psychological effects of unemployment [58], it might also be useful to target individual differences, such as resilience, and to develop tendencies to cope with stress in a highly adaptive manner. Training intervention designed to promote resilient coping might have the effect of increasing individuals’ positive feelings of self-worth and thus reduce their susceptibility to the negative social and psychological effects of unemployment (e.g., depressive symptoms). Also, our findings might have practical utility for mental health and career counseling services with unemployed individuals. EI, resilient coping and levels of self-esteem might potentially be used as a screening assessment to identify individuals at risk of developing depressive symptoms, providing a target for enhancement through occupational training programs. The development of these emotional resources would contribute to the increased mental health and well-being of this population, which might have positive effects on the willingness to actively seek or re-enter paid employment [59].

This study has some limitations. First, it used a cross-sectional design which precludes any causal inference. Future studies could employ longitudinal designs to explore the directionality between the study variables among the unemployed. Accordingly, although our statistical model suggests that resilient coping precedes self-esteem, the possibility remains that there are alternative (potentially bidirectional) causal associations implied. Second, our study encompassed a community sample of unemployed adults, so results might not be generalizable to the clinical population. Future studies testing this serial mediation model in clinical samples would be useful. Third, our study only used self-reported data, which depend on participants’ perceptions and therefore may lead to reporting biases, such as those related to common method variance and social desirability. Finally, the current study constitutes an exploratory research on the mechanisms acting in sequence in the relationship between EI and depression among unemployed. We used a global measure of EI and did not test for the specific contribution of sociodemographic data in the mediation. Further studies should explore whether certain dimensions of EI can better explain this link and investigate the contribution of sociodemographic variables on EI, self-esteem, resilient coping and depressive symptoms among unemployed. The results of such studies could contribute to strengthening the findings of the current research. Relatedly, due to the total explained variance, it is suggested that, beyond resilient coping and self-esteem, other potential mediating factors should also be considered in future studies. For example, some studies have shown that work-role centrality, some personal resources (i.e., core-self evaluations) and social support networks could act as protective factors during unemployment [6]. Thus, it would be useful to include these dimensions in future studies to determine whether they might add further explanatory power in predicting psychological functioning outcomes in unemployed people.

Despite these limitations, these findings provide new insights into the associations between EI and depression among the unemployed population, which may advance a coherent and multifaceted theoretical framework on the pathways through which depression among the unemployed may be prevented or reduced through EI.

## 5. Conclusions

Our results offer new insights into how emotional competences are negatively associated with depressive symptoms in unemployed people. Particularly, resilience and self-esteem have been found to mediate in sequence the relationship between EI and depression. Given the high rate of mental health problems among this population and its detrimental effects on both their well-being and employability, it would be relevant for employment counselors and practitioners to incorporate emotional competences assessment and training in preventive and therapeutic programming.

## Figures and Tables

**Figure 1 ijerph-18-00697-f001:**
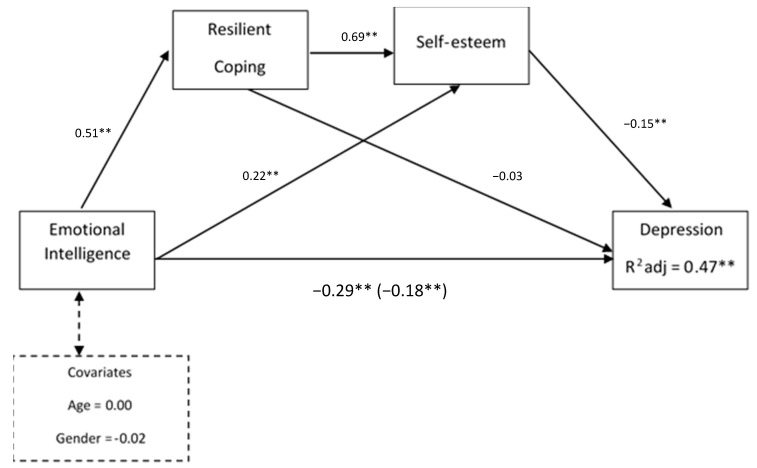
Multiple mediation model for the effect of EI on depressive symptoms via resilient coping and self-esteem controlling for age and gender as covariates. Notes: total effect (c-path) is given in parentheses; ** *p* < 0.01

**Table 1 ijerph-18-00697-t001:** Means, standard deviations, reliabilities and correlations of the variables of interest.

	1	2	3	4
1. Emotional Intelligence	-			
2. Resilient coping	0.61 **	-		
3. Self-esteem	0.49 **	0.60 **	-	
4. Depression	−0.35 **	−0.30 **	−0.36 **	-
M	5.41	3.68	3.40	0.86
SD	0.98	0.82	1.16	0.80
Alpha	0.92	0.81	-	0.91

Note: ** *p* < 0.01.

**Table 2 ijerph-18-00697-t002:** Testing the pathways of the serial mediation model.

Mediation Analysis Path	*b*	95% Bias-Corrected CI	
		Lower	Upper
Total effect	−0.29 ^a^	−0.356	−0.220
Direct effect	−0.18 ^a^	−0.263	−0.094
Total Indirect effect	−0.10 ^a^	−0.172	−0.055
EI→resilience→depression	−0.01	−0.083	0.043
EI→self-esteem→Depression	−0.05 ^a^	−0.089	−0.029
EI→resilience→self-esteem→depression	−0.03 ^a^	−0.068	−0.015
Model F (5, 500) = 20.54; *p* < 0.001; R = 0.41; R^2^ adj = 0.17			

Note: All paths were estimated while controlling for gender and age; standardized regression coefficients shown for each path. EI = emotional intelligence ^a^ Empirical 95% confidence interval does not include zero.

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
