# Peer review of "A Sequential Path Model Testing: Emotional Intelligence, Resilient Coping and Self-Esteem as Predictors of Depressive Symptoms during Unemployment"

_ijerph, 2021, doi:10.3390/ijerph18020697_

Round 1
Reviewer 1 Report
The article is of clinical and social interest. The introduction is well founded. The sample is enough. The methodology and data analysis are correct.
The sociodemographic data would be better if the confidential interval of the mean (95%) was given for age and time in months of unemployment.
In paragraph 3.2. The Serial Mediation Analysis on line 168 says: "To examine whether the relationship between EI and depression was mediated sequentially by resilient coping and self-esteem, the SPSS macro PROCESS by Hayes (2013)." The 2013 Hayes citation has no number and does not appear in the bibliographic references
The discussion does not mention the possible relationship between educational levels and time of unemployment with depression, resilience and self-esteem and emotional intelligence.
Can there be different relationships for people with more time unemployed or with a lower educational level?
Author Response
Please, see the attachment

Reviewer 2 Report
The authors propose a study on the negative effects of being unemployed. In particular, the study investigates the mechanisms that can explain the positive effects of EI in preventing the onset of depressive symptoms due to unemployment.
The topic addressed by the authors is highly relevant, given the current European context. Moreover, the concepts under study are well introduced, and the methodological properties described appear robust. The final discussion concerning practical and professional implications is also fascinating.
I consider the article well structured and clear, with some minor revisions to be considered prior to publication, listed below.
- In the abstract, I would introduce the concept of EI before proposing the mediation mechanisms.
The introduction is well done, but it is not entirely clear why the authors decided to focus exclusively on self-esteem and resilient coping and not on other dimensions related to EI. - In the description of the dimensions analysed, it would be useful to bring examples of items for each of them. I would pay attention to the italics; in some cases, it is present in others not.
- It would be useful to explain more in detail the rationale of the statistical choices chosen. For example, why was a bootstrap option chosen?
- It would be useful to review the practical implications and theoretical impact also in function of gender and age related results, which should not be underestimated. Are there any considerations from these points of view? Also, given the characteristics of the sample and the level of education, are there any considerations in this respect as well?
- Linked to a previous point, the variance explained by the model suggests that other variables may be introduced in the future, any suggestions for future research?
Reviewer 3 Report
The basic idea of the inquiry is highly significant. The article is very well framed and the research process is presented in systematic and scientific way. I would, however, like to raise some points that may be considered to further improve the quality of the manuscript.
Title:
- The present title of the article is a bit lengthy, therefore it is recommended to put double dots after testing, such as “A sequential path model testing: Emotional intelligence, resilient coping and self-esteem as predictors of depressive symptoms during unemployment”.
Introduction:
- The introduction of the article is well-formed and well written. However, in the start authors stated that, “The unemployment rate has dramatically increased in southern European countries in the last two decades, especially after the global economic crisis, affecting particularly Greece (17%), Spain (15.8%) and Italy (9.7%), which manifestly exceed the EU average rate (7.9%) [Eurostat, July 2020] [1]”. Though, a recent published article in the “journal of Health Psychology” stated that the unemployment rate in Greece is 18.3% and Spain 16.2% citing the same report i.e. Eurostat in August 2020. I am wonder that how in a month the unemployment rate changed that much. It is therefore, advised to recheck the mentioned report and figure out the correct unemployment rate, to not confuse the readers. For your ease I am including the said article “Pathways from emotional intelligence to well-being and health outcomes among unemployed: Mediation by health-promoting behaviours”.
- After reviewing the introduction and overall study I came across that the authors have cited most recent articles which is highly appreciable, but as it is the end of 2020, and I found only one reference “39”, which is published in 2020. I assume that present study should present outcomes of the relevant literature review to justify the inquiry providing its theoretical grounds by citing some recent articles which are published in 2020. To support the basic idea of the study. For example:
- Pathways from emotional intelligence to well-being and health outcomes among unemployed: Mediation by health-promoting behaviours.
- The impact of emotional intelligence on depression among international students studying in China: The mediating effect of acculturative stress.
- The effectiveness of emotional intelligence in the face of terrorism fear and employees’ mental health strain.
- The Influence of Emotional Intelligence on Resilience, Test Anxiety, Academic Stress and the Mediterranean Diet. A Study with University Students.
I hope that these studies will highly improve the quality of the manuscript.
- Some references are not as per the journal style. Proof reading may also improve the readability of the manuscript.
Method:
- Method section is overall satisfactory. I however could not understand why did authors not recognize and consider the sub factors of the emotional intelligence scale. As these factors represent different aspects of the respective constructs presented by the authors of scales. Inclusion of these factors separately in the further data analysis may provide evidence regarding separate contribution of each aspect in influencing the mediators and DV. Moreover, I advised to provide two or three items from the given scale would be great, such like the scale of Resilient Coping, Self-esteem and depressive symptoms to make the study more consistent.
- I found that method section of this study has much similarities with a recent published article in the area. It is therefore, advised to work on the possible plagiarism issue in this section. For your ease I am including the said article “Pathways from emotional intelligence to well-being and health outcomes among unemployed: Mediation by health-promoting behaviours”.
Results:
- This study has used self-reported data, which may cause social desirability bias. Therefore, it is request to perform the Conman Method Bias (CMB) test and report its results.
Discussion:
- In line 216-218, as mentioned that Resilient coping didn't mediated the relationship between EI and depression, please provide arguments in support of these results. That what are the reasons of not supporting these relationships.
Minor Issues:
Except these major issues, following are some of the minor points which may need to be considered for the improvement of the article.
- In line 43 the authors stated, “chronic diseases impair unemployed individuals´ ability to word, and increase the likelihood of drifting into long-term unemployment”, here what does the ability to word means please make it sample for the readers.
- In line 57 the authors talk about, “lower negative affect [16,18,19]), better subjective well-being please make sure the use of small bracket here, this may be a typing mistake.
- Line 60, suggesting that IE might be a protective factor against depression [13, 23]. I think authors are talking about EI not IE.
- Please make sure that in line 75,76 and 78 the citation style is correct, also it would be great if these types of mistakes are omitted in the whole manuscript.
- Line 128, it would be better to mention the acronym of Brief Resilient Coping Scale (BRCS) over here.
- In line 169, the authors stated that “5,000 bootstrap samples was used”, Please provide rational of using bootstrap samples in your study.
- Line 263-264 of the conclusions, “emotional competences are negatively associated with depressive symptoms unemployed”. “emotional competences are negatively associated with depressive symptoms in unemployed”. I think “in” is required.
Best of Luck
Reviewer 4 Report
Introduction
- As a suggestion, they could include a most up-to-date quote from the last 5 years.
- Regarding the first section, it is suggested that they write the hypotheses of their study in a more clarifying way.
Method
- In relation to the participants section, you must indicate the percentage of the sample of men, not only women.
- The authors should add the data analysis section, thus specifying why they have used the different tests in the results.
Discussion
1. The discussion is a comparison between its results and those found by other cited authors, previously in the introduction. Therefore, it is observed that all studies that appear in the discussion do not appear in the introduction and must be present in the introduction.
Author Response
Please, see the attachment

Round 2
Reviewer 3 Report
The present form of the paper has been improved significantly as compare to the previous version. Therefore, i am going to accept the paper in current form just one point to be address that is:
In line 253 of the new submitted version of the manuscript, "That is, while EI had a significant direct effect", It is advised to recheck the given sentence as both "That is" and "While" does not make any sense here. It would be better to remove That is.